# Tryptophan Hydroxylase 2 Deficiency Modifies the Effects of Fluoxetine and Pargyline on the Behavior, 5-HT- and BDNF-Systems in the Brain of Zebrafish (*Danio rerio*)

**DOI:** 10.3390/ijms222312851

**Published:** 2021-11-27

**Authors:** Valentina S. Evsiukova, Daria Bazovkina, Ekaterina Bazhenova, Elizabeth A. Kulikova, Alexander V. Kulikov

**Affiliations:** 1Department of Psychoneuropharmacology, Federal Research Center Institute of Cytology and Genetic Siberian Branch of Russian Academy of Sciences, 630090 Novosibirsk, Russia; evsiukova@bionet.nsc.ru (V.S.E.); kulikova@bionet.nsc.ru (E.A.K.); 2Department of Behavioral Neurogenomics, Federal Research Center Institute of Cytology and Genetic Siberian Branch of Russian Academy of Sciences, 630090 Novosibirsk, Russia; daryabazovkina@gmail.com; 3Department of Genetic Collections of Neural Disorders, Federal Research Center Institute of Cytology and Genetic Siberian Branch of Russian Academy of Sciences, 630090 Novosibirsk, Russia; ekaterina.yu.bazhenova@gmail.com

**Keywords:** tryptophan hydroxylase 2, serotonin, BDNF, brain, behavior, antidepressant resistance, translational study, zebrafish

## Abstract

The mechanisms of resistance to antidepressant drugs is a key and still unresolved problem of psychopharmacology. Serotonin (5-HT) and brain-derived neurotrophic factor (BDNF) play a key role in the therapeutic effect of many antidepressants. Tryptophan hydroxylase 2 (TPH2) is the rate-limiting enzyme in 5-HT synthesis in the brain. We used zebrafish (*Danio rerio*) as a promising model organism in order to elucidate the effect of TPH2 deficiency caused by p-chlorophenylalanine (pCPA) on the alterations in behavior and expression of 5-HT-related (*Tph2*, *Slc6a4b*, *Mao*, *Htr1aa*, *Htr2aa*) and BDNF-related (*Creb*, *Bdnf*, *Ntrk2a*, *Ngfra*) genes in the brain after prolonged treatment with two antidepressants, inhibitors of 5-HT reuptake (fluoxetine) and oxidation (pargyline). In one experiment, zebrafish were treated for 72 h with 0.2 mg/L fluoxetine, 2 mg/L pCPA, or the drugs combination. In another experiment, zebrafish were treated for 72 h with 0.5 mg/L pargyline, 2 mg/L pCPA, or the drugs combination. Behavior was studied in the novel tank diving test, mRNA levels were assayed by qPCR, 5-HT and its metabolite concentrations were measured by HPLC. The effects of interaction between pCPA and the drugs on zebrafish behavior were observed: pCPA attenuated “surface dwelling” induced by the drugs. Fluoxetine decreased mRNA levels of *Tph2* and *Htr2aa* genes, while pargyline decreased mRNA levels of *Slc6a4b* and *Htr1aa* genes. Pargyline reduced *Creb*, *Bdnf* and *Ntrk2a* genes mRNA concentration only in the zebrafish treated with pCPA. The results show that the disruption of the TPH2 function can cause a refractory to antidepressant treatment.

## 1. Introduction

Depressive disorders are among the leading causes of mental disability in industrial countries [1,2,3]. Selective serotonin reuptake inhibitors (SSRIs) occupy the leading position on the global antidepressant drug market [4]. They block the serotonin (5-HT) transporter and, thereby increase 5-HT concentration in the synaptic cleft [5,6,7,8,9]. Unfortunately, about 40% of depressive patients remain refractory to antidepressant treatment [10,11,12]. According to the neurotrophic hypothesis, the therapeutic effect of SSRIs is mediated by activation of cAMP-response protein (CREB) and synthesis of the brain derived neurotrophic factor (BDNF) [13,14]. At the same time, prolonged SSRI treatment affects the brain’s 5-HT homeostasis, including its synthesis, release, reception, reuptake, and oxidation [15], as well as decreases the 5-HT level in mouse brains [16,17].

Tryptophan hydroxylase 2 (TPH2) is the key enzyme of 5-HT synthesis in the brain [18,19]. TPH2 irreversible inhibitor, p-chlorophenylalanine (pCPA) [20,21] as well as *Tph2* gene knockout [22,23,24] dramatically reduce the 5-HT level in mouse brains. TPH2 involvement in the mechanism of resistance to antidepressant drug treatment is hypothesized [25]. However, the results of experimental studies on the effect of TPH2 deficiency on response to antidepressants are contradictory. Thus, pCPA-induced [26] and hereditary TPH2 deficiency attenuated the behavioral response in the forced swim test to acute treatment with 5-HT reuptake inhibitors, citalopram, and paroxetine, in mice [26,27,28]. However, other authors did not confirm these results [29].

The chronic administration of SSRIs to Tph2KI mice with hereditary TPH2 deficiency results in a dramatic drop in brain 5-HT levels [17]. It was hypothesized that chronic administration of inhibitors of the key enzymes of 5-HT oxidation, monoamine oxidases A and B, those that increase the 5-HT level, could be more effective than SSRIs in the treatment of individuals with hereditary TPH2 deficiency [30].

The zebrafish (*Danio rerio*) is a promising model species for clarifying the role of TPH2 in the behavioral and neurochemical responses to chronic stimulation of 5-HT neurotransmission due to molecular homology of the brain 5-HT system in zebrafish and mammals [31,32]. Moreover, zebrafish can be treated with drugs via aquarium water. This treatment mode is not stressful. The TPH2 activity and the 5-HT level in the zebrafish brain can be easily reduced by pCPA [33]. In addition, zebrafish demonstrate a specific “surface dwelling” response to SSRI treatment [34,35,36,37].

Here we used zebrafish as a model organism in order to elucidate the influence of TPH2 deficiency on the effects of prolonged treatment with SSRI (fluoxetine) and MAO inhibitor (pargyline) on behavior, 5-HT- and BDNF-systems in the brain. We intended to compare the behavior in the novel tank diving test, the levels of 5-HT and its main metabolite, 5-hydroxyindoleacetic acid (5-HIAA), concentrations of mRNA of *Tph2*, *Slc6a4b* (5-HT transporter), *Mao* (monoamine oxidase), *Htr1aa* (5-HT_1A_ receptor), *Htr2aa* (5-HT_2A_ receptor), *Creb*, *Bdnf*, *Ntrk2a* (TrkB receptor), *Ngfra* (p75 receptor) genes in the brain of control zebrafish and those treated for 72 h with pCPA, fluoxetine, pargyline or these drugs combinations.

## 2. Results

### 2.1. Effects of pCPA and Fluoxetine on Zebrafish Behavior in the Novel Tank Diving Test

The effect of the “Fluoxetine” factor, but not the “pCPA” factor or the factor’s interaction on locomotor activity (distance traveled), was shown (Table 1). The Tukey’s *post hoc* test showed that fluoxetine treatment significantly decreased the distance traveled, while pCPA did not affect this trait (Figure 1).

Significant effects of the “pCPA” × “Fluoxetine” factors interaction on distance from the tank’s bottom, time spent in the lower and upper thirds of tanks were revealed (Table 1). The Tukey’s post hoc comparisons did not reveal a significant difference in these traits between pCPA-treated and control groups (Figure 1). At the same time, fluoxetine decreased the time spent in the lower third, and increased time spent in the upper third and distance from the tank’s bottom, thereby fluoxetine increased “surface dwelling” (Figure 1). pCPA significantly attenuated the effects of fluoxetine on the studied characteristics (Figure 1).

### 2.2. Effects of pCPA and Fluoxetine on 5-HT Metabolism in Zebrafish Brain

The remarkable effects of the “pCPA” and “Fluoxetine” factors, but not their interaction on 5-HT and 5-HIAA levels in the zebrafish brain, were shown (Table 2). Both drugs similarly decreased 5-HT and 5-HIAA levels in the brain (Figure 2). At the same time, a significant effect of the interaction of the “pCPA” and “Fluoxetine” factors on the 5-HIAA/5-HT ratio was found: the combination of these drugs caused about a 3-fold increase in the ratio (Figure 2). It could be hypothesized that during the combined treatment with pCPA and fluoxetine the effects of these drugs were summarized resulting in a more intensive reduction of the 5-HT level.

### 2.3. Effects of pCPA and Fluoxetine on mRNA Levels of Tph2, Slc6a4b, Mao, Htr1aa, Htr2aa, Creb, Bdnf, Ntrk2a and Ngfra Genes in Zebrafish Brain

Although the significant effect of the “Fluoxetine” factor on mRNA levels of *Tph2*, *Slc6a4b*, *Mao*, *Htr2aa*, *Ntrk2a* genes was revealed (Table 3), the *post hoc* Tukey’s test revealed only a significant decrease in *Tph2* and *Htr2aa* gene mRNA levels in the zebrafish brain treated with fluoxetine and fluoxetine together with pCPA compared to that of the control group (Figure 3). No effect of fluoxetine on the mRNA levels of *Htr1aa*, *Creb*, *Bdnf*, and *Ngfra* genes in zebrafish brain, as well as those of the “pCPA” factor and the “pCPA” × “Fluoxetine” interaction on the variability of the mRNA levels of all studied genes, was shown (Table 3, Figure 3).

### 2.4. Effects of pCPA and Pargyline on Zebrafish Behavior in the Novel Tank Diving Test

The significant effects of the “Pargyline” × “pCPA” interaction on distance traveled (locomotor activity), distance from the tank’s bottom, time spent in the lower and upper parts of the tank in zebrafish (Table 4, Figure 4) were shown (Table 4).

The Tukey’s *post hoc* comparisons did not reveal a significant difference in these traits between pCPA-treated and control groups (Figure 4). Pargyline decreased traveled distance but increased “surface dwelling” by decreasing time spent in the lower third, and increasing time spent in the upper third and distance from the tank’s bottom. pCPA attenuated the effects of pargyline on these behavior features (Figure 4).

### 2.5. Effects of pCPA and Pargyline on 5-HT Metabolism in Zebrafish Brain

The marked effect of the “pCPA” and “Pargyline” interaction on 5-HT and 5-HIAA levels, but not on the 5-HIAA/5-HT ratio was shown (Table 5). pCPA attenuated the pargyline-induced rise in the 5-HT level without any effect on the pargyline-induced decrease in the 5-HIAA level and the 5-HIAA/5-HT ratio (Figure 5).

The significant effect of the “Pargyline” factor on 5-HT, 5-HIAA levels, and the 5-HIAA/5-HT ratio in the zebrafish brain was observed (Table 5). Pargyline dramatically increased the 5-HT level and decreased 5-HIAA concentration as well as the 5-HIAA/5-HT ratio in the brain. (Figure 5).

In this experiment, the Tukey’s *post hoc* test did not reveal an expected decrease of 5-HT in the brains of zebrafish treated with pCPA. The ANOVA analysis revealed that the brain 5-HT level in this experiment is defined by “pCPA”, “Pargyline” factors and their interaction. The effect of the “Pargyline” factor on the brain 5-HT level was the highest (Table 5) and it masked an expected decrease of 5-HT level in the brain of zebrafish treated with pCPA.

### 2.6. Effects of pCPA and Pargyline on the mRNA Levels of Tph2, Slc6a4b, Mao, Htr1aa, Htr2aa, Creb, Bdnf, Ntrk2a and Ngfra Genes in Zebrafish Brain

No effect of the “pCPA” and “Pargyline” interaction on the studied genes’ mRNA levels was observed (Table 6). Although significant effects of the “pCPA” factor on *Mao*, *Htr2aa*, *Bdnf,* and *Ngfra* genes expression, as well as the “Pargyline” factor on the level of *Slc6a4b*, *Htr1aa*, *Creb*, *Bdnf,* and *Ntrk2a* gene mRNAs, were revealed (Table 6), the Tukey’s *post hoc* analysis revealed the decrease in *Slc6a4b*, *Mao*, *Htr1aa*, *Htr2aa*, *Bdnf*, *Ntrk2a, and Ngfra* genes mRNAs only in the group treated with pargyline together with pCPA (Figure 6).

## 3. Discussion

The main idea of the present study was to use zebrafish as a model organism in order to compare the effects of prolonged treatment with SSRI fluoxetine and MAO inhibitor pargyline on behavior, brain 5-HT and BDNF systems in zebrafish with pCPA-induced TPH2 deficiency. There were four main reasons for the choice of zebrafish: (1) molecular homology in the brain 5-HT system between zebrafish and mammalians [31,32]; (2) they can be treated chronically and without stress via aquarium water; (3) rearing of zebrafish in water containing 2 mg/L of pCPA significantly reduces the 5-HT level in their brain [33]; (4) SSRIs cause-specific “surface dwelling” behavior in the novel tank diving test [34,35,36,37]. It should be emphasized that this “surface dwelling” seems to be a specific response to SSRIs treatment as well as a decrease in immobility time in the forced swim test in laboratory rodents.

We found that the exposition of zebrafish for 72 h to 2 mg/L of TPH2 inhibitor, pCPA, significantly reduced the 5-HT level in their brain. Thereby, the selected pCPA concentration, protocol, and time of exposition are effective and sufficient to inhibit TPH2 and it is in good concordance with our previous results [33].

As expected, prolonged treatment with pargyline and fluoxetine produces opposite effects on the 5-HT level in the brain. Pargyline inhibited MAO and dramatically increased the 5-HT level in the zebrafish brain. This result agrees with that previously observed earlier in zebrafish [33]. Earlier we showed that an acute (for 3 h) treatment with fluoxetine did not affect the 5-HT level in the brain of zebrafish [38]. Now we found that prolonged treatment with fluoxetine decreased 5-HT and 5-HIAA levels in the brain of zebrafish. This finding agrees with those observed in mice chronically treated with fluoxetine [16,17] and seems to result from a disruption of the mechanism of 5-HT reutilization in the brain [25]. However, in mice this fluoxetine-induced decrease in the 5-HT level is observed after 30 [16] or 60 [17] days of treatment, while in zebrafish 3-day-long treatment is sufficient to dramatically reduce the 5-HT level. It seems that the persistent treatment during 72 h by fluoxetine dissolved in aquarium water accelerates the fluoxetine-induced drop of 5-HT level in the zebrafish brain.

In this study for the first time, we revealed the principal difference in the effects of pargyline and fluoxetine on the brain 5-HT level and metabolism in the pCPA-treated zebrafish. As expected, pCPA attenuated a pargyline-induced increase in the 5-HT level, but it remained higher than in the control group. At the same time, fluoxetine blocking 5-HT recovery further aggravates the decrease in the neurotransmitter level caused by pCPA. The last finding is in good agreement with the fluoxetine-induced dramatic reduction of the 5-HT level in the brain of Tph2KI mice with hereditary TPH2 deficiency [17].

Zebrafish placed in the novel and potentially dangerous tank usually prefer the lower part and seldom visit the surface of the tank [34,35,36,37]. Some authors interpret this reaction (“bottom dwelling”) to novelty as anxiety-related behavior [34,39,40,41]. Treatment of zebrafish with pCPA (2 × 300 mg/kg pCPA, ip) increases anxiety in the novel tank diving test and decreased time spent in the tank’s upper third [37,42]. This result agrees with an increase of anxiety in Tph2KI mice with TPH2 deficiency [43].

Acute [35,37,38] or chronic [34,35] treatment with fluoxetine increases time spent near the tank’s surface (“surface dwelling”) in zebrafish. Recently we showed that acute treatment with imipramine also induced “surface dwelling” in zebrafish [36]. Here we revealed “surface dwelling” behavior in zebrafish treated with pargyline for 72 h.

Some authors interpret “surface dwelling” as a demonstration of “serotonin syndrome’—specific behavioral response to an increase in extracellular 5-HT concentration [44]. However, other authors interpret this antidepressant-induced “surface dwelling” as an anxiolytic effect of the drug [34,37]. It needs to emphasize that chronic treatment with fluoxetine also decreases anxiety in mice in the open field test [45,46].

In experiments with rodents when drugs were injected *ip* or *per os*, prolonged or chronic treatment usually covers time windows of several weeks. Although we used a relatively short administration of the drugs for 72 h, the observed alterations in behavior and in 5-HT concentration in the brain of zebrafish resembled those resulting from chronic treatment. It seems that the persistent treatment for 72 h by these drugs added in aquarium water accelerates their effects on zebrafish brain and behavior.

One of the most intriguing results of our study is the significant effects of “pCPA” × “Fluoxetine” and “pCPA” × “Pargyline” interactions on the time spent in the lower and upper thirds as well as the mean distance from the tank’s bottom. We found that pCPA attenuated the effects of fluoxetine and pargyline on these traits. This result indicates that TPH2 activity is important for “surface dwelling” expression induced by treatment with fluoxetine and pargyline.

The mechanism of behavioral effects of joint administration of pCPA and pargyline is clear: pCPA decreases 5-HT synthesis and attenuates the pargyline-induced increase in the brain neurotransmitter concentration thereby reducing 5-HT-induced “surface dwelling” expression and restoring the drop of locomotion. At the same time, the mechanism of behavioral effect of combined pCPA and fluoxetine administration is more questionable. We showed that the combined administration of these drugs dramatically reduced the 5-HT level in zebrafish brain more than pCPA and fluoxetine separately and, therefore, we might expect a drop of time spent in the upper third down to the levels observed in control and pCPA-treated zebrafish. Instead, we observed a paradoxical increase in time spent in the upper part of the tank in zebrafish treated with fluoxetine together with pCPA compared to control animals. Two possible explications of this paradox can be proposed. The first, despite a dramatic reduction of the total 5-HT level in the brain, the fluoxetine-induced increase in the 5-HT level in the synaptic cleft remains sufficient to increase time spent in the upper third. The second, fluoxetine itself can activate the BDNF system. Indeed, fluoxetine can bind to the TrkB receptor transmembrane domain and facilitate its activation by BDNF [47].

A pragmatic question arises: is there a relationship between the fluoxetine- and pargyline-induced “surface dwelling” in the novel tank diving test and the therapeutic effect of these drugs. On one hand, there is no doubt the close link between an antidepressant-induced increase in the 5-HT level in the synaptic cleft and the therapeutic effect of these antidepressants. On the other hand, some authors associate the antidepressant-induced “surface dwelling” with the increase in the 5-HT level in the synaptic cleft (“5-HT syndrome”) [44]. Therefore, the interpretation of “surface dwelling” as a behavioral index of the 5-HT system activation in zebrafish may be a useful predictor of potential antidepressant activity.

The brain’s 5-HT and BDNF systems play a key role in the therapeutic effects of SSRI [14,15]. Moreover, prolonged SSRI treatment affects the brain’s 5-HT homeostasis, including its synthesis, release, reception, reuptake, and oxidation [15], as well as decreases the 5-HT level in mouse brains [16,17]. Thereby, the second step of our study was the evaluation of fluoxetine and pargyline effects on mRNA levels of 5-HT- and BDNF-related genes in the zebrafish brain. Zebrafish genes *Tph2*, *Slc6a4b*, *Htr1aa*, *Htr2aa*, *Creb*, *Bdnf*, *Ntrk2a* and *Ngfra* are homologs of mammalian genes *Tph2*, *Slc6a4*, *Htr1a*, *Htr2a*, *Creb*, *Bdnf*, *Ntrk2* and *Ngfr*. However, in contrast to mammals, the only gene *Mao* encodes enzyme MAO in zebrafish. In order to minimize the risk of false positives, we used the Tukey’s test for *post hoc* comparisons.

The Tukey’s test revealed that fluoxetine concentration of 0.2 mg/L significantly decreased *Tph2* and *Htr2aa* genes expression in the whole brain of adult zebrafish: the levels of these mRNA in zebrafish brain treated with fluoxetine and fluoxetine together with pCPA were reduced compared to those of the control zebrafish. Although ANOVA showed the effect of the “Fluoxetine” factor also on the mRNA levels of *Slc6a4b* and *Mao* genes, the influence of this drug seemed too weak and it was not confirmed by the Tukey’s test. Earlier it was shown, that the acute treatment with high doses of fluoxetine (5 mg/L, 2 h) did not affect *Htr1aa* and *Slc6a4* gene expression in adult zebrafish brain [48]. However, subchronic (during 80 h) treatment of zebrafish larvae with different doses of fluoxetine reveal no effect of the drug on *Mao* gene expression, while fluoxetine in 0.8 µM concentration increased the *Htr1aa* mRNA level and in 0.0015 and 0.5 µM decreased *Slc6a4* gene expression [49]. This difference between these and our data can be explained by the difference in age of animals and concentration of the drug.

The Tukey’s test revealed a decrease in the mRNA levels of *Slc6a4b*, *Mao*, *Htr1aa*, *Htr2aa* genes in zebrafish brains treated with pargyline together with pCPA compared to the control ones. At the same time, unlike fluoxetine, the pargyline treatment did not affect *Tph2* gene expression. These results indicate the involvement of the *Htr2aa* gene in the mechanism of both drugs’ action and agree with the commonly accepted phenomenon of these receptors’ down-regulation by antidepressant drugs [50,51,52].

The close interaction between the brain BDNF and 5-HT systems is beyond doubt [53,54]. Acute treatment with a high (5 mg/L, 2 h) concentration of fluoxetine increased *Bdnf* gene expression in the zebrafish brain [48]. There was no information on the effect of fluoxetine on *Creb* gene expression as well as on *Ntrk2a* and *Ngfra* genes coding receptors for BDNF and pro-BDNF in the zebrafish brain.

In our experiment, the Tukey’s test did not reveal any difference in the mRNA levels of *Creb*, *Bdnf*, *Ntrk2a,* and *Ngfra* genes in zebrafish treated with fluoxetine and fluoxetine together with pCPA compared with the control ones. These results agree with data obtained earlier on rodents [55,56,57,58,59,60,61,62] and zebrafish [48] and indicate that the BDNF system in zebrafish is resistant to prolonged fluoxetine treatment.

At the same time, the Tukey’s test revealed a remarkable decrease in the levels of *Bdnf* and *Ntrk2a* genes in the brain of pargyline × pCPA treated zebrafish compared with the control ones. At the same time, pargyline itself did not affect these genes expression. It is commonly accepted, that activation of the brain BDNF system plays the key role in the therapeutic effect of antidepressants [13,14]. Therefore, the observed decrease in these BDNF-related genes expression in the brain in zebrafish treated with pargyline together with pCPA may be interpreted as an adverse effect of pargyline treatment on the brain BDNF-system in TPH2 deficient individuals.

There are molecular and pharmacological homologies between zebrafish and rodents. First, enzymes of metabolism, transporter, and receptors of 5-HT in zebrafish have their mammalian homologs. Second, in zebrafish and mammals, pCPA, fluoxetine, and pargyline inhibit TPH2, 5-HT transporter, and MAO, respectively. Third, in zebrafish and mice, chronic fluoxetine treatment produces an anxiolytic effect in the novel tank diving [34,35] and the open field [45,46] tests, respectively. Therefore, the present study carried out on zebrafish reveals the key role of TPH2 in the response to antidepressant treatment and provides experimental evidence that the disruption of the TPH2 function can cause a refractory to antidepressant treatment.

## 4. Materials and Methods

### 4.1. Zebrafish

The study was conducted in the Department of Genetic Collections of Neural Disorders at the Federal Research Centre Institute of Cytology and Genetics, Siberian Branch of the Russian Academy of Sciences in accordance with the recommendations of the Guidelines for the use of zebrafish in the NIH intramural research program of 12 April 2013, and was approved by the Committee on the Ethics of Animal Experiments of the Russian National Center of Genetic Resources of Laboratory Animals (protocol No. 34 of 15 June 2016). All efforts were made to minimize the number of zebrafish and their suffering.

The experiments were carried out on 160 six-month-old males and females (1:1) of leopard strain zebrafish. This strain was selected due to its high anxiety [63]. Zebrafish were bred in the Collective Centre of Animal Genetic Resources (supported by the basic research project No 0259-2021-0015). From the age of one month, the fish were reared in two mixed groups of 80 males and 80 females (1:1) in two 125-L glass tanks equipped with CX-300 bacterial filters (Chosen, China) at temperature 27 °C and artificial photoperiod 12 h light and 12 h dark, with daybreak at 9:00 a.m. The fish were fed two times per day 6 days per week with Tetramin Tropical Flakes (Tetra, Blacksburg, VA, USA) and one day per week with frozen blood worms (*Chironomus plumosus*).

### 4.2. Drugs and Treatments

Fluoxetine, (±)-N-methyl-γ-[4-(trifluoromethyl) phenoxy] benzenepropanamine hydrochloride (Merck, Darmstadt, Germany); pargyline, N-methyl-N-(2-propynyl)benzylamine hydrochloride (Merck, Darmstadt, Germany) and pCPA, (R)-2-amino-3-phenylpropionic acid (Merck, Darmstadt, Germany) were added to tank water in the final concentrations of 0.2, 0.5 mg/L and 2 mg/L, respectively. These concentrations were considered effective based on our earlier experiments of 72 h treatment [33,35]. To evaluate the effect of TPH2 inhibition on the response to fluoxetine and pargyline we performed two separate experiments.

Experiment 1. Here we evaluate the influence of pCPA-induced TPH2 deficiency on SSRI effects. Eighty fish were divided into 4 experimental groups with 20 fish in each: (1) control (clear water), (2) pCPA (2 mg/L of pCPA), (3) fluoxerine (0.2 mg/L of fluoxetine) and (4) pCPA + fluoxetine (2 mg/L of pCPA and 0.2 mg/L of fluoxetine).

Experiment 2. Here we evaluate the influence of pCPA-induced TPH2 deficiency on MAO inhibitor effects. Eighty fish were divided into 4 experimental groups with 20 fish in each: (1) control (clear water), (2) pCPA (2 mg/L of pCPA), (3) pargyline (0.5 mg/L of pargyline) and (4) pCPA + pargyline (2 mg/L of pCPA and 0.5 mg/L of pargyline).

Zebrafish of these experimental groups were reared over 72 h in well-aerated 8 L glass tanks (10 fish per tank) filled with clear water or drug solutions. Every day at 5:00 p.m. 90% water or drug solutions were replaced with fresh water or solutions, correspondently. The fish were fed with blood worms at 10:00 a.m. After 72 h of treatment, half of the fish from each tank were tested in the novel tank diving test and then euthanized, while another half of the fish were euthanized without testing in order to reduce the experiment duration. The animals were euthanized by immersion into cold water (+2 °C). Whole brains of fish were immediately removed, frozen with liquid nitrogen, and stored at −80 °C. Finally, 20 brains from each group were randomly divided into two blocks of 10 brains each: one block for 5-HT assay and another block for mRNA extraction.

### 4.3. The Novel Tank Diving (NT) Test

The behavioral response to novelty was tested in 10 zebrafish from each group. Ten animals are sufficient for statistical analysis of behavior [34,37]. The test was conducted in the daytime (11:00 a.m.–1:00 p.m.) in a glass test tank (24 cm in length, 15 cm in depth, and 7 cm in width, Figure 7A) according to the protocol that was described in earlier studies [35]. The fish was individually transferred from its home tank into the test tank and then the tank together with the fish inside was placed into the apparatus and the recording started immediately. The fish position was automatically recorded for 5 min with the rate of 30 fps by a Web camera connected to a computer via a USB 2.0 port. The stream of frames was automatically analyzed in real-time by the EthoStudio software and saved on a hard disk as a compressed video file. The EthoStudio frame by frame separated the pixels associated with the fish from those associated with the background applying the threshold algorithm [35,64] and calculating the coordinates of the fish center (Figure 7B). Moreover, the EthoStudio automatically calculated the density map corresponding to the distribution of fish-associated pixels in the tank [35,64] (Figure 7C). The sequence of coordinates and the density map were used to evaluate (1) the distance traveled (m), (2) the mean distance from the tank’s bottom (cm), the time spent (%) in (3) the lower and (4) the upper thirds of the tank. The third and fourth parameters were calculated as the ratios of fish-associated pixels sum in these parts of the tank to that in the whole tank [35,64]. The test tank was washed after each test. The researcher was blind to the groups.

### 4.4. 5-HT and 5-HIAA Concentrations Assay by HPLC

The brain was homogenized in 100 µL of cold 0.6 M HClO_4_ using a motor-driven grinder (Z359971, Sigma-Aldrich, Saint-Louis, MO, USA) and the homogenate was spun for 15 min at 12,000 rpm (+4 °C). The pellet was diluted in 1 mL of 0.1 M NaCl and used for protein quantitation by Bradford (Bio Rad, Hercules, CA, USA) according to the manufacturer’s protocol. The clear supernatant was diluted twice with pure water and used for assay of 5-HT and 5-HIAA by HPLC on Luna C18 (2) column (5 μM particle size, L × I.D. 75 × 4.6 mm, Phenomenex, Torrance, CA, USA) with electrochemical detection (750 mV, DECADE II™ Electrochemical Detector; Antec, Industrieweg, The Netherlands) as it was described in previous studies [33,65]. The standard mixes containing 1, 2, and 3 ng of serotonin (5-HT) and 5-hydroxyindoleacetic acid (5-HIAA) were repeatedly assayed throughout the entire procedure and used to plot the calibration curves for each substance. The areas of peaks were estimated using LabSolution LG/GC software version 5.54 (Shimadzu Corporation, Duisburg, Germany) and calibrated against the calibrated curves for corresponding standards [33,65]. The contents of 5-HT and 5-HIAA were expressed in ng/mg protein assayed by Bradford.

### 4.5. mRNA Level Assay by qPCR

The brain was homogenized in 300 µL of Trizol reagent (Bio Rad, Hercules, CA, USA) using a motor-driven grinder (Z359971, Sigma-Aldrich, Saint-Louis, MO, USA). Total mRNA extraction; RNA treatment with RNAase free DNAase (Promega, Maison, WI, USA); cDNA synthesis with a random hexanucleotide primer and R01 Kit (Biolabmix, Novosibirsk, Russia), and SYBR Green real-time quantitative PCR with selective primers (Table 7) and R401 Kit (Sintol, Moscow, Russia) were performed in accordance with the protocols of the manufacturers. As external standards, we used solutions containing 25, 50, 100, 200, 400, 800, 1600, 3200, and 6400 copies of genomic DNA extracted from zebrafish muscles. The gene expression was presented as a relative number of cDNA copies calculated on 100 copies of *Polr2eb* cDNA as an internal standard [33,65].

### 4.6. Statistics

All data were tested using the Kolmogorov’s test and met the assumption of normality. The data were presented as the mean ± SEM, analyzed with two-way ANOVA with “pCPA” and “Fluoxetine” (experiment 1) or “Pargyline” (experiment 2) as independent factors. The factors’ interaction was also calculated. *Post hoc* analyses were carried out using the Tukey’s HSD multiple comparison test as appropriate. Statistical significance was set at *p* < 0.05.

## 5. Conclusions

Here we tried to solve a fundamental problem of the relationship between antidepressant efficacy and TPH2 activity. First, we showed that pharmacological reduction of this enzyme activity attenuated the behavioral response to prolonged treatments with inhibitors of 5-HT reuptake (fluoxetine) and MAO (pargyline). Second, prolonged fluoxetine treatment dramatically drops the brain 5-HT level in individuals with TPH2 deficiency. Third, prolonged pargyline treatment resulted in a rise of the brain’s 5-HT level even in zebrafish treated with pCPA. Fourth, prolonged pargyline treatment produced an adverse effect on the brain BDNF system in zebrafish with TPH2 deficiency. Thereby, our study did not show any advantage of MAO inhibitors in comparison to SSRI in zebrafish with TPH2 deficiency. Moreover, disruption of the TPH2 function can be a possible reason for refractory to antidepressant treatment. This factor should be taken into account when prescribing antidepressants to patients.

## Figures and Tables

**Figure 1 ijms-22-12851-f001:**
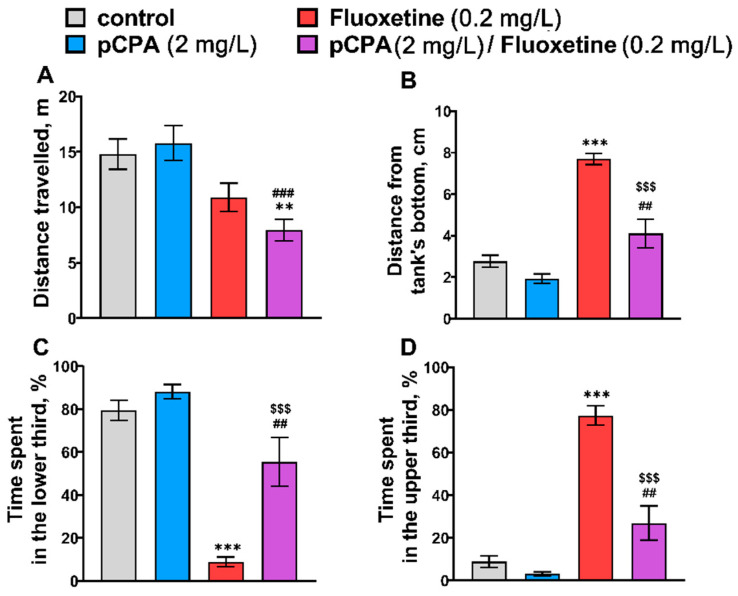
(**A**) Distance traveled (m), (**B**) mean distance from the tank’s bottom (cm), time (%) (**C**) spent in the lower and (**D**) the upper thirds in the novel tank diving test in control (water) zebrafish and those exposed for 72 h to pCPA (2 mg/L), fluoxetine (0.2 mg/L) or combination of pCPA (2 mg/L) and fluoxetine (0.2 mg/L), correspondently. The number of animals in each group was 10. ** *p* < 0.01, *** *p* < 0.001 vs. control group; ^##^ *p* < 0.01, ^###^ *p* < 0.001 vs. pCPA treated group; ^$$$^ *p* < 0.001 vs. fluoxetine treated group.

**Figure 2 ijms-22-12851-f002:**
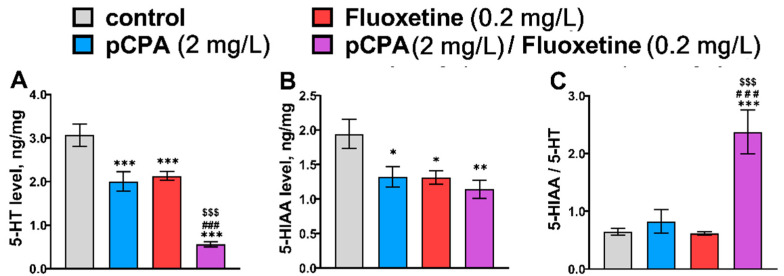
Concentration of (**A**) 5-HT (ng/mg of protein), (**B**) 5-HIAA (ng/mg of protein) and (**C**) 5-HIAA/5-HT ratio in the brain of control (water) zebrafish and those exposed for 72 h to pCPA (2 mg/L), fluoxetine (0.2 mg/L) or combination of pCPA (2 mg/L) and fluoxetine (0.2 mg/L), correspondently. The number of animals in each group was 10. * *p* < 0.05, ** *p* < 0.01, *** *p* < 0.001 vs. control group; ^###^
*p* < 0.001 vs. pCPA treated group; ^$$$^ *p* < 0.001 vs. fluoxetine treated group.

**Figure 3 ijms-22-12851-f003:**
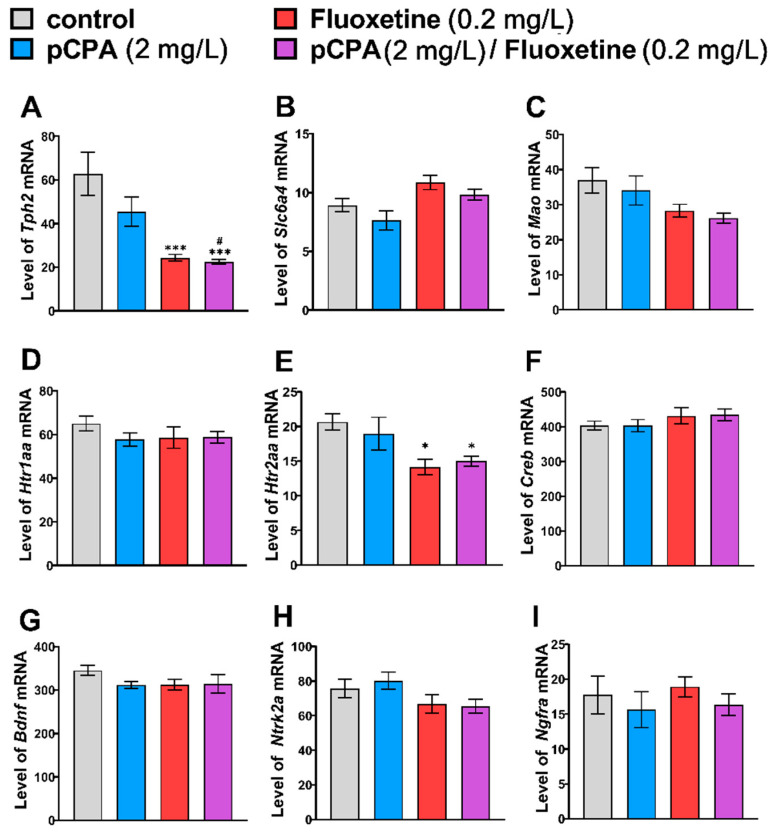
Levels of mRNA of (**A**) *Tph2*, (**B**) *Slc6a4b*, (**C**) *Mao*, (**D**) *Htr1aa*, (**E**) *Htr2aa*, (**F**) *Creb*, (**G**) *Bdnf*, (**H**) *Ntrk2a* and (**I**) *Ngfra* genes in the brain of control (water) zebrafish and those exposed for 72 h to pCPA (2 mg/L), fluoxetine (0.2 mg/L) or combination of pCPA (2 mg/L) and fluoxetine (0.2 mg/L), correspondently. The number of animals in each group was 10. * *p* < 0.05, *** *p* < 0.001 vs. control group; ^#^
*p* < 0.05 vs. pCPA treated group.

**Figure 4 ijms-22-12851-f004:**
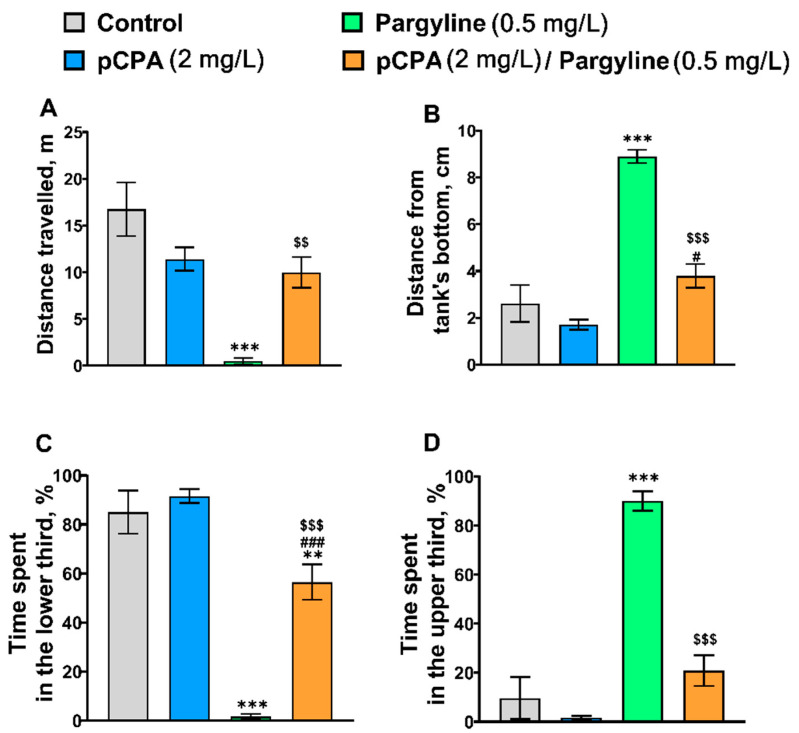
(**A**) Distance traveled (m), (**B**) mean distance from the tank’s bottom (cm), (**C**) time (%) spent in the lower and (**D**) the upper thirds in the novel tank diving test in control (water) zebrafish and those exposed for 72 h to pCPA (2 mg/L), pargyline (0.5 mg/L) or combination of pCPA (2 mg/L) and pargyline (0.5 mg/L), correspondently. The number of animals in each group was 10. ** *p* < 0.01; *** *p* < 0.001 vs. control group; ^#^ *p* < 0.05, ^###^ *p* < 0.001 vs. pCPA treated group; ^$$^ *p* < 0.01, ^$$$^ *p* < 0.001 vs. pargyline treated group.

**Figure 5 ijms-22-12851-f005:**
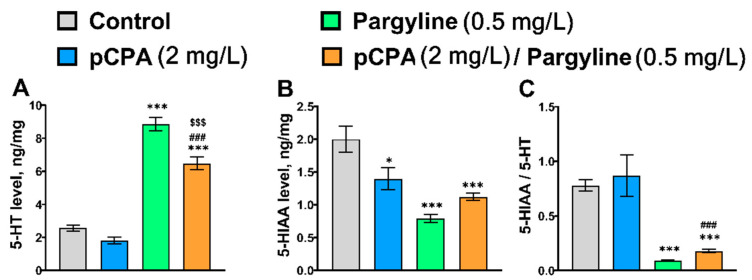
Concentration of (**A**) 5-HT (ng/mg of protein), (**B**) 5-HIAA (ng/mg of protein) and (**C**) 5-HIAA/5-HT ratio in the whole brain of control (water) zebrafish and those exposed for 72 h to pCPA (2 mg/L), pargyline (0.5 mg/L) or combination of pCPA (2 mg/L) and pargyline (0.5 mg/L), correspondently. The number of animals in each group was 10. * *p* < 0.05, *** *p* < 0.001 vs. control group; ^###^ *p* < 0.001 vs. pCPA treated group; ^$$$^ *p* < 0.001 vs. pargyline treated group.

**Figure 6 ijms-22-12851-f006:**
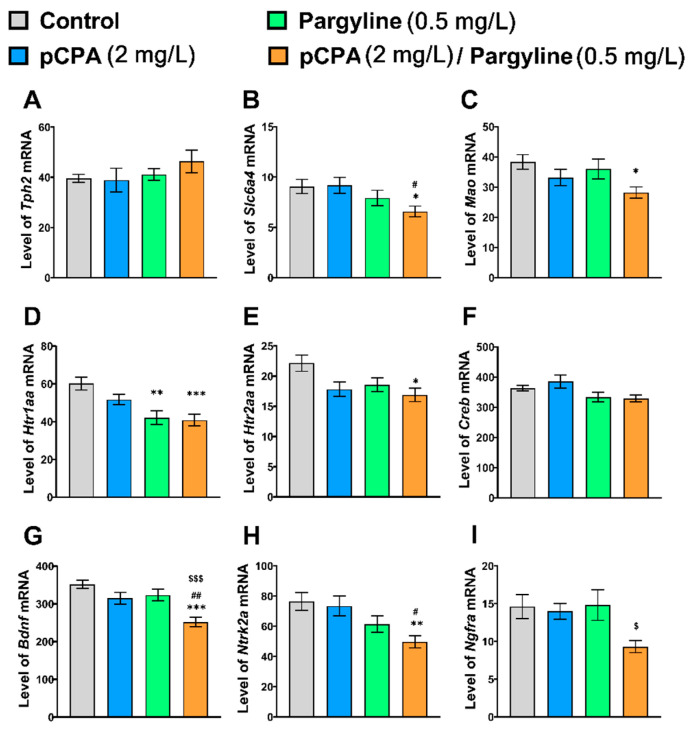
Levels of mRNA of (**A**) *Tph2*, (**B**) *Slc6a4b*, (**C**) *Mao*, (**D**) *Htr1aa*, (**E**) *Htr2aa*, (**F**) *Creb*, (**G**) *Bdnf*, (**H**) *Ntrk2a* and (**I**) *Ngfra* genes in the whole brain of control (water) zebrafish and those exposed for 72 h to pCPA (2 mg/l), pargyline (0.5 mg/L) or combination of pCPA (2 mg/L) and pargyline (0.5 mg/L), correspondently. The number of animals in each group was 10. * *p* < 0.05, ** *p* < 0.01, *** *p* < 0.001 vs. control group; ^#^ *p* < 0.05, ^##^ *p* < 0.01 vs. pCPA treated group; ^$^ *p* < 0.05, ^$$$^ *p* < 0.001 vs. pargyline treated group.

**Figure 7 ijms-22-12851-f007:**
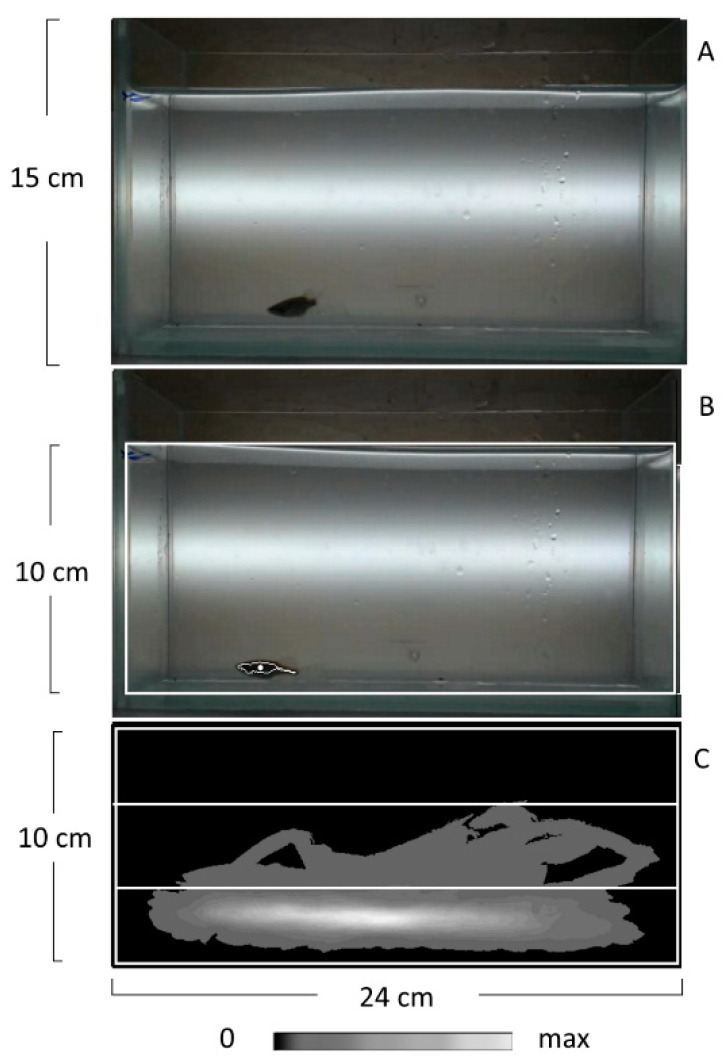
Tracking zebrafish in the novel tank diving test by EthoStudio software. (**A**) Photo of the tank for testing (24 cm in length, 15 cm in depth, and 7 cm in width) filled with water and with a zebrafish inside. (**B**) The white rectangle marks the water surface selected as the tracking arena, which is 24 cm in length and 10 cm in depth. EthoStudio software finds the contour and the center of a zebrafish and marks them with a white line and a point, respectively. (**C**) EthoStudio software calculates the density map of pixel distribution associated with zebrafish on the arena. Pixel density is coded gray according to the scale placed below the map. The arena is divided into three equal thirds. The map shows that most of the pixels associated with a zebrafish are in the lower third.

**Table 1 ijms-22-12851-t001:** Two-way ANOVA of the effect of “pCPA”, “Fluoxetine” factors and their interaction on the variability of distance traveled, mean distance from tank’s bottom, time spent in the lower third, time spent in the upper third of the tank in the novel tank diving test in zebrafish.

Trait	pCPA (2 mg/L)	Fluoxetine (0.2 mg/L)	Interaction
Distance traveled	F_1,36_ < 1	**F_1,36_ = 20.0, *p* < 0.001**	F_1,36_ = 2.3, *p* = 0.14
Distance from tank’s bottom	**F_1,36_ = 28.0, *p* < 0.001**	**F_1,36_ = 72.5, *p* < 0.001**	**F_1,36_ = 10.9, *p* = 0.002**
Time spent in the lower third	**F_1,36_ = 18.4, *p* < 0.001**	**F_1,36_ = 64.0, *p* < 0.001**	**F_1,36_ = 8.6, *p* = 0.006**
Time spent in the upper third	**F_1,36_ = 34.1, *p* < 0.001**	**F_1,36_ = 92.1, *p* < 0.001**	**F_1,36_ = 21.5, *p* < 0.001**

Statistically significant values are marked in bold.

**Table 2 ijms-22-12851-t002:** Two-way ANOVA of the effect of “pCPA”, “Fluoxetine” factors and their interaction on the variability of levels of 5-HT, 5-HIAA and 5-HIAA/5-HT ratio in the brain of zebrafish.

Trait	pCPA (2 mg/L)	Fluoxetine (0.2 mg/L)	Interaction
5-HT	**F_1,36_ = 54.2, *p* < 0.001**	**F_1,36_ = 44.2, *p* < 0.001**	F_1,36_ = 2.0, *p* = 0.17
5-HIAA	**F_1,36_ = 6.5, *p* = 0.015**	**F_1,36_ = 6.9, *p* = 0.013**	F_1,36_ = 2.1, *p* = 0.16
5-HIAA/5-HT	**F_1,36_ = 18.4, *p* < 0.001**	**F_1,36_ = 11.3, *p* = 0.002**	**F_1,36_ = 12.2, *p* = 0.0013**

Statistically significant values are marked in bold.

**Table 3 ijms-22-12851-t003:** Two-way ANOVA of the effect of “pCPA”, “Fluoxetine” factors and their interaction on the variability of mRNA level of *Tph2*, *Slc6a4b*, *Mao*, *Htr1aa*, *Htr2aa*, *Creb*, *Bdnf*, *Ntrk2a*, *Ngfra* genes in the brain of zebrafish.

Gene	pCPA (2 mg/L)	Fluoxetine (0.2 mg/L)	Interaction
*Tph2*	F_1,36_ = 2.7, *p* = 0.11	**F_1,36_ = 27.0, *p* < 0.001**	F_1,36_ = 1.7, *p* = 0.20
*Slc6a4b*	F_1,36_ = 3.5, *p* < 0.07	**F_1,36_ = 10.9, *p* = 0.002**	F_1,36_ < 1
*Mao*	F_1,36_ < 1	**F_1,36_ = 7.8, *p* = 0.008**	F_1,36_ < 1
*Htr1aa*	F_1,36_ < 1	F_1,36_ < 1	F_1,36_ = 1.0, *p* = 0.32
*Htr2aa*	F_1,36_ < 1	**F_1,36_ = 12.7, *p* < 0.001**	F_1,36_ < 1
*Creb*	F_1,36_ < 1	F_1,36_ 2.6, *p* = 0.12	F_1,36_ < 1
*Bdnf*	F_1,36_ = 1.9, *p* = 0.26	F_1,36_ = 1.2, *p* = 0.28	F_1,36_ = 1.6, *p* = 0.22
*Ntrk2a*	F_1,36_ < 1	**F_1,36_ = 5.9, *p* = 0.02**	F_1,36_ < 1
*Ngfra*	F_1,36_ = 1.2, *p* = 0.28	F_1,36_ < 1	F_1,36_ < 1

Statistically significant values are marked in bold.

**Table 4 ijms-22-12851-t004:** Two-way ANOVA of the effect of “pCPA”, “Pargyline” factors and their interaction on the variability of distance traveled, mean distance from tank’s bottom, time spent in the lower third, time spent in the upper third of the tank in the novel tank diving test in zebrafish.

Trait	pCPA (2 mg/L)	Pargyline (0.5 mg/L)	Interaction
Distance traveled	F_1,36_ = 1.3, *p* = 0.26	**F_1,36_ = 23.4, *p* < 0.001**	**F_1,36_ = 16.5, *p* < 0.001**
Distance from tank’s bottom	**F_1,36_ = 34.6, *p* < 0.001**	**F_1,36_ = 67.1, *p* < 0.001**	**F_1,36_ = 16.8, *p* = 0.002**
Time spent in the lower third	**F_1,36_ = 25.8, *p* < 0.001**	**F_1,36_ = 96.4, *p* < 0.001**	**F_1,36_ = 15.9, *p* = 0.006**
Time spent in the upper third	**F_1,36_ = 44.7, *p* < 0.001**	**F_1,36_ = 74.5, *p* < 0.001**	**F_1,36_ = 28.1, *p* < 0.001**

Statistically significant values are marked in bold.

**Table 5 ijms-22-12851-t005:** Two-way ANOVA of the effect of “pCPA”, “Pargyline” factors and their interaction on the variability of levels of 5-HT, 5-HIAA, and 5-HIAA/5-HT ratio in the brain of zebrafish.

Trait	pCPA (2 mg/L)	Pargyline (0.5 mg/L)	Interaction
5-HT	**F_1,36_ = 24.3, *p* < 0.001**	**F_1,36_ = 299.6, *p* < 0.001**	**F_1,36_ = 6.7, *p* = 0.014**
5-HIAA	F_1,36_ = 1.03, *p* = 0.32	**F_1,36_ = 31.0, *p* = 0.015**	**F_1,36_ = 12.2, *p* = 0.0013**
5-HIAA/5-HT	F_1,36_ < 1	**F_1,36_ = 57.0, *p* < 0.001**	F_1,36_ < 1

Statistically significant values are marked in bold.

**Table 6 ijms-22-12851-t006:** Two-way ANOVA of the effect of “pCPA”, “Pargyline” factors and their interaction on the variability of mRNA level of *Tph2*, *Slc6a4b*, *Mao*, *Htr1aa*, *Htr2aa*, *Creb*, *Bdnf*, *Ntrk2a*, *Ngfra* genes in the brain of zebrafish.

Gene	pCPA (2 mg/L)	Pargyline (0.5 mg/L)	Interaction
*Tph2*	F_1,36_ < 1	F_1,36_ = 1.6, *p* = 0.22	F_1,36_ < 1
*Slc6a4b*	F_1,36_ < 1	**F_1,36_ = 7.0, *p* = 0.012**	F_1,36_ = 1.0, *p* = 0.32
*Mao*	**F_1,36_ = 6.2, *p* = 0.018**	F_1,36_ = 1.9, *p* = 0.26	F_1,36_ < 1
*Htr1aa*	F_1,36_ = 2.2, *p* = 0.14	**F_1,36_ = 19.4, *p* < 0.001**	F_1,36_ = 1.2, *p* = 0.28
*Htr2aa*	**F_1,36_ = 6.2, *p* = 0.018**	F_1,36_ = 3.6, *p* = 0.07	F_1,36_ = 1.3, *p* = 0.27
*Creb*	F_1,36_ < 1	**F_1,36_ = 7.9, *p* = 0.008**	F_1,36_ < 1
*Bdnf*	**F_1,36_ = 15.6, *p* < 0.001**	**F_1,36_ = 11.3, *p* = 0.002**	F_1,36_ = 1.6, *p* = 0.22
*Ntrk2a*	F_1,36_ = 1.8, *p* = 0.19	**F_1,36_ = 12.4, *p* < 0.001**	F_1,36_ < 1
*Ngfra*	**F_1,36_ = 4.4, *p* = 0.042**	F_1,36_ = 2.4, *p* = 0.13	F_1,36_ = 2.8, *p* = 0.10

Statistically significant values are marked in bold.

**Table 7 ijms-22-12851-t007:** Sequences, annealing temperatures of the primers, and size of PCR products (amplicons).

Gene	Primer Sequences	Annealing Temperatures, °C	Amplicon Size, bp
Polr2eb	F5′-GTGACGCAGGATGAATTGGA-3′ R5′-CACCAGGACTGTCAGGTCATT-3′	62	105
Tph2	F5′-TCTACTACAACCCTTACACGCAGA-3′ R5′-CGTCACAGACGGTGGTTAAG-3′	62	105
Slc6a4b	F5′-ACCGCAAATCCAATGACCGAT-3′ R5′-CGCTCACGGGAACCTCTG-3′	63	144
Mao	F5′-AAACCATGCACTTGATGACTGA-3′ R5′-TCATACTTGCCATACCCCCTG-3′	62	121
Hr1aa	F5′-GCTGCACTTCTTCCATCCTG-3′ R5′-GGTTTCCTCCAACCCAACAT-3′	61	178
Htr2aa	F5′-TTTGGCAGTGGTTTGTGAAC-3′ R5′-ATCCAGTGAGTGGCAGGTGT-3′	61	257
Creb	F5′-GCTTTGAATCGCAGACATCA-3′ R5′-ATGGCATAATCGTGGTCGTT-3′	60	409
Bdnf	F5′-TGCGAGTTATAGTGCCGCTT-3′ R5′-AGCCGCCGTTACTCTTTCTC-3′	63	313
Ntrk2a	F5′-TATTCCCTTCAGCGTGTCTGG-3′ R5′-GCATGAAATGAGCAGATACGG-3′	62	233
Ngfra	F5′-GATTTAGATCGTCTGTGGAGC-3′ R5′-AAAATGATGTACGCCAGGAG-3′	59	161

## Data Availability

Not applicable.

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
