# Peer review of "Tryptophan Hydroxylase 2 Deficiency Modifies the Effects of Fluoxetine and Pargyline on the Behavior, 5-HT- and BDNF-Systems in the Brain of Zebrafish (Danio rerio)"

_ijms, 2021, doi:10.3390/ijms222312851_

Round 1

Reviewer 1 Report

In this work the authors propose zebrafish (Danio rerio) as an ideal experimental model for investigating the effects of antidepressant compounds (the SSRI Fluoxetine and the MAO inhibitor Pargyline) on the associated anxiety-motor behavior, as well as on serotonin (5-HT) and BDNF pathways and associated metabolism; more specifically, animals with pCPA-induced TPH2 deficiency are used.

The authors chose zebrafish as experimental model for three main reasons supported by previous research: the high homology in the brain 5-HT system between zebrafish and mammalians, the pCPA-dependent reduction of 5-HT level in the fish brain, the SSRIs induced “surface dwelling” behavior in the novel tank diving test.

The structure of the manuscript is overall well constructed and sufficiently clear in identifying the answers to the experimental questions that the authors pose as the goal of the research.

The sections of the text are also well balanced and comprehensive.

The experimental design is well organized and developed by an interesting (although not really innovative) approach, especially regarding the use of the “novel tank diving test” in the context of several (single and combined) pharmacological treatments.

(Main concerns:)

Anyway, I'm not entirely sure I fully understand the final interpretation that the authors provide of the data produced. I would ask them to clarify more in deep some points in the text:

  1. what is the physiological significance of the pCPA treatment? Based on the fact that this inhibitor of the specific enzime TPH2 induces a reduction of its activity, and as a consequence a decrease in the physiological levels of serotonine, I interpret this approach as aimed at artificially reproducing a condition of stress/depression in the animal: am I correct? In that case, it should be stressed the point more clearly in the text.
  2. it is recognized that some locomotor behaviors have specific physiological correlates in fish: for example, among the different researches, Rupert et al, (doi: 1016/j.bbr.2009.06.022 ) already assessed zebrafish as a model of stress and anxiety by examining how behavioral and physiological phenotypes are affected by various environmental, pharmacological and genetic factors, and used the novel tank approach to demonstrate that “fluoxetine had robust anxiolytic effects, including increased exploration and reduced erratic movements”. ……..it would be nice that authors add this reference in the text. A general physiological interpretation of all locomotor behaviors analysed (distance travelled, distance from tank’s bottom, time spent in the lower/upper third) should be added in the Intro/discussion sections as well to facilitate the comprehension of the whole research context by the readers which are not neccessarly confident with behavioral tests in laboratory animals, and in fish specifically.
  3. An other important point is the interpretation of short(acute) and prolonged (chronic) treatment: I wonder whether the 72h of drugs administration adopted by the authors can really be considered a prolonged treatment; typically, prolonged treatments cover time windows of several weeks, or even months, especially when  recordings are programmed of behavioral parameters associated with anxiety /depressive states: the authors should motivate and support the choice of their 72-h administration protocol intended as prolonged treatment (research cited  from literature, i.e. #17, applied longer administration times, indeed). This is a critical point to clarify, considering that based on data in literature acute treatment with antidepressant (i.e. Fluoxetin) seem to promote an increase of 5-HT in the synaptic cleft, whereas a prolonged (chronic) administration of this drug leads to a progressive decrease of 5-HT in experimental animals. In order to asses that 72h have the effect of a chronic treatment in zebrafish in the context of pCPA-induced “depressive” condition”, it should be eventually compared with an acute treatment (few hours): I strongly suggest the authors to add this experiment, analysing the effect of a short-time antidepressant drug (fluoxetine) treatment on a group of fish ( treated versus control group), to confirm that zebrafish responds in the similarly to other experimental models with a 5-HT increase or decrease, under acute or chronic antidepressant treatment respectively, supporting the authors choise of the 72h time windows as a prolonged treatment for this model.

(minor concerns:)

  • A minor suggestion that I consider useful to make the work easier to read and follow conceptually, would be the creation of a table with the list of all the acronyms used in this research: the authors have correctly entered each term in full, introducing appropriately the acronyms, but any unfamiliar readers of the various protein factors and categories of pharmacological compounds adopted in the study could find the continuous search for the definition of an acronym through the text difficult: a table would facilitate a quick consultation and make the reading of the work more fluent.
  • Line 160 (paragraph 2.5): in figure 5 I observe a significant variation (decrease, of pCPA treatment only in 5-HIAA metabolite, whether in the text the authors punctualize that the only effect of the pCPA treatment was observed in the 5_HT levels: can you explain the discrepancy between graphic data and text interpretation? Are the two factor simply inverted in the text?
  • Lines 164-65 and 188-89: what the authors mean with the sentences “This discrepancy in the effect of pCPA on 5-HT level in the experiments 1 and 2 results from an increase in the variabilities of these traits in the experiment 2” and “This discrepancy in the effect of pCPA on genes expression between the experiments 1 and 2 results from an increase in the variabilities of these traits in the experiment 1”. Are the authors referring to high standard deviations? Please clarify.
  • Line 279 (discussion): replace “using” with “used”

Overall, the paper would be suitable for publication in the journal, respecting its criteria and research areas. The data are clearly presented, the experimental methods explained in full detail. However, in my opinion, I find that the discussion is missing in clarifying what the final purpose of the work is, and what actual novelty it brings in the field of physiological and pharmacological studies in the use of antidepressants:  it is not clear to me whether the main purpose of the work is to demonstrate and attest a simple interaction between multiple factors (TPH2, Serotonin and BNDF pathways), or rather to highlight/discuss the functional role and therapeutic validity of some antidepressants in a context of "induced stress", or again to support the experimental use of zebrafish as a suitable model in pharmacological studies aimed at clarifying the function and therapeutic activity of these drugs.

I suggest the author to givein the discussion/conclusions a clearer explanation of the take home message that can be extrapolated from the set of experimental data presented.

In any case, I would suggest the paper for publication only in the case that all the points above addressed will be explained and discussed in the text in more details

Author Response

Answer to reviewer 1

In this work the authors propose zebrafish (Danio rerio) as an ideal experimental model for investigating the effects of antidepressant compounds (the SSRI Fluoxetine and the MAO inhibitor Pargyline) on the associated anxiety-motor behavior, as well as on serotonin (5-HT) and BDNF pathways and associated metabolism; more specifically, animals with pCPA-induced TPH2 deficiency are used.

The authors chose zebrafish as experimental model for three main reasons supported by previous research: the high homology in the brain 5-HT system between zebrafish and mammalians, the pCPA-dependent reduction of 5-HT level in the fish brain, the SSRIs induced “surface dwelling” behavior in the novel tank diving test.

The structure of the manuscript is overall well constructed and sufficiently clear in identifying the answers to the experimental questions that the authors pose as the goal of the research.

The sections of the text are also well balanced and comprehensive.

The experimental design is well organized and developed by an interesting (although not really innovative) approach, especially regarding the use of the “novel tank diving test” in the context of several (single and combined) pharmacological treatments.

(Main concerns:)

Anyway, I'm not entirely sure I fully understand the final interpretation that the authors provide of the data produced. I would ask them to clarify more in deep some points in the text:

  1. what is the physiological significance of the pCPA treatment? Based on the fact that this inhibitor of the specific enzime TPH2 induces a reduction of its activity, and as a consequence a decrease in the physiological levels of serotonine, I interpret this approach as aimed at artificially reproducing a condition of stress/depression in the animal: am I correct? In that case, it should be stressed the point more clearly in the text.

Answer: We used pCPA treatment to model TPH2 and 5-HT deficiency. The contradictory data concerning the behaviors effects of TPH2 deficiency in mice were discussed in our review [30]. Two references [43] on an anxiogenic effect of pCPA in zebrafish [42] and hereditary TPH2 deficiency in mice [43] was added and the corresponding item of the discussion was rewritten (lines 233-238).

  1. it is recognized that some locomotor behaviors have specific physiological correlates in fish: for example, among the different researches, Rupert et al, (doi: 1016/j.bbr.2009.06.022 ) already assessed zebrafish as a model of stress and anxiety by examining how behavioral and physiological phenotypes are affected by various environmental, pharmacological and genetic factors, and used the novel tank approach to demonstrate that “fluoxetine had robust anxiolytic effects, including increased exploration and reduced erratic movements”. ……..it would be nice that authors add this reference in the text. A general physiological interpretation of all locomotor behaviors analysed (distance travelled, distance from tank’s bottom, time spent in the lower/upper third) should be added in the Intro/discussion sections as well to facilitate the comprehension of the whole research context by the readers which are not neccessarly confident with behavioral tests in laboratory animals, and in fish specifically.

Answer: We cited Egan’s publication [34] several times in the Introduction and Discussion. We provided a detailed analysis of the antidepressant-induced “surface dwelling” behavior in zebrafish and compare it with anxiolytic effect of fluoxetine in the open field test in mice (lines 239-247). Two references [45,46] concerning an anxiolytic effect of fluoxetine in the open field test were added.

  1. An other important point is the interpretation of short(acute) and prolonged (chronic) treatment: I wonder whether the 72h of drugs administration adopted by the authors can really be considered a prolonged treatment; typically, prolonged treatments cover time windows of several weeks, or even months, especially when  recordings are programmed of behavioral parameters associated with anxiety /depressive states: the authors should motivate and support the choice of their 72-h administration protocol intended as prolonged treatment (research cited  from literature, i.e. #17, applied longer administration times, indeed). This is a critical point to clarify, considering that based on data in literature acute treatment with antidepressant (i.e. Fluoxetin) seem to promote an increase of 5-HT in the synaptic cleft, whereas a prolonged (chronic) administration of this drug leads to a progressive decrease of 5-HT in experimental animals. In order to asses that 72h have the effect of a chronic treatment in zebrafish in the context of pCPA-induced “depressive” condition”, it should be eventually compared with an acute treatment (few hours): I strongly suggest the authors to add this experiment, analysing the effect of a short-time antidepressant drug (fluoxetine) treatment on a group of fish ( treated versus control group), to confirm that zebrafish responds in the similarly to other experimental models with a 5-HT increase or decrease, under acute or chronic antidepressant treatment respectively, supporting the authors choise of the 72h time windows as a prolonged treatment for this model.

Answer: Earlier we showed that an acute (3 h) treatment with 0.25 mg\l of fluoxetine failed to affect 5-HT level in the zebrafish brain (lines 215-216) [38]. In other publication [35] we showed that alterations in the behavior of zebrafish in the novel tank diving test induced by acute (3 h) fluoxetine administration less than those induced by chronic (14 days) fluoxetine treatment. In the present study we found that 72 h treatment with fluoxetine and  pargyline produced dramatic alterations in 5-HT level and metabolism and behavior similar to those of chronic treatment. That is why we consider 72 h treatment zebrafish with these drugs as prolonged (not short or acute). We think that the permanent treatment for 72 h by the drugs dissolved in water produced similar strong effect of the brain and behavior of zebrafish as chronic (for weeks) ip or po treatment in rodents (lines 222-224, 248-253).

(minor concerns:)

  • A minor suggestion that I consider useful to make the work easier to read and follow conceptually, would be the creation of a table with the list of all the acronyms used in this research: the authors have correctly entered each term in full, introducing appropriately the acronyms, but any unfamiliar readers of the various protein factors and categories of pharmacological compounds adopted in the study could find the continuous search for the definition of an acronym through the text difficult: a table would facilitate a quick consultation and make the reading of the work more fluent.
  • Answer: We used only commonly accepted acronyms and defined them in the text. We think that once more table (Table 8) increase the volume but does not facilitate reading of the manuscript.
  • Line 160 (paragraph 2.5): in figure 5 I observe a significant variation (decrease, of pCPA treatment only in 5-HIAA metabolite, whether in the text the authors punctualize that the only effect of the pCPA treatment was observed in the 5-HT levels: can you explain the discrepancy between graphic data and text interpretation? Are the two factor simply inverted in the text?
  • Answer: Significant effects of on 5-HT and 5-HIAA levels in the brain were revealed. Two way ANOVA This post-hoc effect of pCPA on 5-HIAA level results from significant effect of the pCPA x Pargyline interaction on this trait. We did not observed any statistically significant effect of the pCPA factor because the effect of this drug on 5-HIAA was strongly dependent on pargyline treatment: pCPA per se decreased and at the same time attenuated the pargyline-induced decrease of 5-HIAA. At the same time, the brain 5-HT level in this experiments is defined by “pCPA”, “Pargyline” factors and their interaction. The effect of the “Pargyline” factor on the brain 5-HT level was the highest and it masked an expected decrease of 5-HT level in the brain of zebrafish treated with pCPA (lines 164-169).
  • Lines 164-65 and 188-89: what the authors mean with the sentences “This discrepancy in the effect of pCPA on 5-HT level in the experiments 1 and 2 results from an increase in the variabilities of these traits in the experiment 2” and “This discrepancy in the effect of pCPA on genes expression between the experiments 1 and 2 results from an increase in the variabilities of these traits in the experiment 1”. Are the authors referring to high standard deviations? Please clarify.
  • Answer: We agree that these phrases are obscure. The first phrase was clarified (lines 164-169). The second phrase was removed since in both experiments the Tukey’s test did not revealed any alteration in these gene expression in the pCPA-treated group.
  • Line 279 (discussion): replace “using” with “used”
  • Answer: This phrase disappeared after reorganization of the Discussion section.

Overall, the paper would be suitable for publication in the journal, respecting its criteria and research areas. The data are clearly presented, the experimental methods explained in full detail. However, in my opinion, I find that the discussion is missing in clarifying what the final purpose of the work is, and what actual novelty it brings in the field of physiological and pharmacological studies in the use of antidepressants:  it is not clear to me whether the main purpose of the work is to demonstrate and attest a simple interaction between multiple factors (TPH2, Serotonin and BNDF pathways), or rather to highlight/discuss the functional role and therapeutic validity of some antidepressants in a context of "induced stress", or again to support the experimental use of zebrafish as a suitable model in pharmacological studies aimed at clarifying the function and therapeutic activity of these drugs.

Answer: The Discussion section was rewritten according to the reviewer’s recommendations.

I suggest the author to givein the discussion/conclusions a clearer explanation of the take home message that can be extrapolated from the set of experimental data presented.

In any case, I would suggest the paper for publication only in the case that all the points above addressed will be explained and discussed in the text in more details

In general: We corrected some phrases in the Introduction, rewrote the items 2.4, 2.5, 2.6 in the Results and the Discussion sections according to the reviewer recommendations. We also added 5 new references and corrected our English. We hope that our corrections made the manuscript more clear.

We thank the reviewer very much for his\her valuable recommendation.

Reviewer 2 Report

The paper presented seems methodologically sound, impressive even. I have no issues with design or how data is presented; clear and pedagogical figures and tables, and a good disposition. There are some minor, but consistent issues with the language (e.g. definite articles missing or being used where it is unidiomatic) but overall well-written and not too verbose.

I do however have some issues with the context given and, perhaps most fundamentally, there's, at least to my mind, not a clear rationale as to why the study has been undertaken. There are some references to the clinical situation, and a discussion about treatment resistance, but I don't really understand what the authors specifically hoped to elucidate in terms of affective disorders in humans, and no explanation as to why the chosen model organism, chosen behavioural test and chosen pharmacological manipulations would be the best way to do this. 

Apparently the authors are of the opinion that there's an added value of using zebrafish, but is that in relation to mammals? In terms of logistics and cost certainly, but scientifically? I would say that it's hard to argue that, at least in terms of the 5-HT system. Apart from some well-known anatomical differences (mainly the presence of hypothalamic serotonergic nuclei in teleosts) there exist some rather significant differences in terms of how zebrafish respond to various serotonergic manipulations as compared to mammals. To start with, serotonin depletion by way of p-CPA is usually associated with an anxiolytic-like and pro-aggressive effect in rodents (e.g. Vergnes et al., 1984; Näslund et al., 2015), whereas the effect not only in zebrafish (Müller et al., 2020) but also in other teleosts (Lorenzi et al., 2009; Paula et al., 2015) is rather anxiogenic-like and anti-aggressive (but see Adams et al., 1996). Also, acute SSRI treatment is both in rodents as well as in humans assocated with an acute increase in anxiolytic-like behaviour (Näslund et al., 2015), which is the opposite of the situation in zebrafish (Maximino et al., 2013; Karakaya et al., 2021; de Melo et al., 2019) and also at least another teleost (McDonald et al., 2011). Findings relating to cortisol responses are also divergent, with acute SRI treatment inducing a decrease in blood cortisol in zebrafish (Sander de Abreu et al., 2014) while an increase is typically seen in humans and other mammals (Hesketh et al., 2005; Nadeem et al, 2004). Also, both in rodents (Näslund et al., 2015) and in humans (Frick et al., 2015), there are reports of an association between high levels of serotonin/high capacity for serotonin release and/or production on one hand and high anxiety-like behaviour or anxiety disorders on the other hand, with treatment response in humans with generalised anxiety disorder being associated with a reduced capacity (Frick et al., 2016). Zebrafish data is divergent also here (strain differences in Maximino et al., 2013 but see Tran et al., 2016). As the authors mention, the acute serotonin elevation seen during early treatment is transient and the long-term effects are unclear; if anything the early rise in synaptic availability coincides with a period where patients experience heightened anxiety (e.g. Targum et al., 1989; Sinclair et al., 2009; Näslund et al., 2017). In short - zebrafish are pretty different from mammals in terms of how they respond to serotonergic interventions and this should be mentioned. I can't see how it would be better to use zebrafish than say, rats or tree shrews. At best it's neutral, but given these differences, the model organism is in itself a limitation of the paper.

There are, as the authors themselves discuss, indications of lowered serotonergic tone/transmission/availability after long-term treatment in rodents, and I don't really understand why inducing a hyposerotonergic state by way of p-CPA necessarily would be a model of treatment-resistant depression, as there are no indications from the clinical situation for that to be the case. Of course, the long-term effects of SSRI administration are unclear and there are a numerous hypotheses of their mode of action beyond the early effects at the serotonergic synapse, hypotheses that only partly overlap. In general, I feel that the discussion is far too focused on the situation in zebrafish for a paper that desires to in some way help in understanding a human phenomenon; "We showed that the combined administration of these drugs dramatically 256 reduced the 5-HT level in zebrafish brain more than pCPA and fluoxetine separately and, 257 therefore, we might expect an increase in “bottom dwelling” and normalization of lo- 258 comotor activity in zebrafish treated with these drugs. Instead we observed a paradoxical 259 decrease in “bottom dwelling” expression and locomotor activity." is instructive in this regard - it is paradoxical in terms of the zebrafish situation as increased 5-HT levels generally associate with less anxiety-like behaviour, but the opposite is the case in rodents as well as humans.

Some shorter notes:

The acute effect of SSRIs in the forced-swim test is unlikely to be antidepressant in nature, considering the weeks needed for a therapeutic effect in humans, and in other tests/paradigms (e.g. chronic unpredictable stress) - see Anyan & Amir, 2018 for a good discussion of the subject.

I would also strongly object to the statement that "According to commonly accepted neurotrophic hypothesis the therapeutic effect 41 of SSRIs is mediated by activation of cAMP-response protein (CREB) and synthesis of the brain derived neurotrophic factor (BDNF)" - it is one of many hypotheses, and it has rather little in terms of clinical/human data to support it (mostly some candidate gene studies as well as sparse and indirect post-mortem data, as far as I know).

There is also the conflation of models of depression and anxiety - it is not clear why the novel-tank diving test (or the scototaxis test) should be regarded as a model of the former state, and some discussion of what state in humans that the authors wish to model would be nice. This of course ties in to my general desire of clearer rationale for the paper.

Author Response

Answer to reviewer 2

The paper presented seems methodologically sound, impressive even. I have no issues with design or how data is presented; clear and pedagogical figures and tables, and a good disposition. There are some minor, but consistent issues with the language (e.g. definite articles missing or being used where it is unidiomatic) but overall well-written and not too verbose.

I do however have some issues with the context given and, perhaps most fundamentally, there's, at least to my mind, not a clear rationale as to why the study has been undertaken. There are some references to the clinical situation, and a discussion about treatment resistance, but I don't really understand what the authors specifically hoped to elucidate in terms of affective disorders in humans, and no explanation as to why the chosen model organism, chosen behavioural test and chosen pharmacological manipulations would be the best way to do this. 

Answer: More arguments explaining the advantages of zebrasfish as model organism were added in the Introduction (lines 59-65) and the Discussion (lines 200-2004).

Apparently the authors are of the opinion that there's an added value of using zebrafish, but is that in relation to mammals? In terms of logistics and cost certainly, but scientifically? I would say that it's hard to argue that, at least in terms of the 5-HT system. Apart from some well-known anatomical differences (mainly the presence of hypothalamic serotonergic nuclei in teleosts) there exist some rather significant differences in terms of how zebrafish respond to various serotonergic manipulations as compared to mammals. To start with, serotonin depletion by way of p-CPA is usually associated with an anxiolytic-like and pro-aggressive effect in rodents (e.g. Vergnes et al., 1984; Näslund et al., 2015), whereas the effect not only in zebrafish (Müller et al., 2020) but also in other teleosts (Lorenzi et al., 2009; Paula et al., 2015) is rather anxiogenic-like and anti-aggressive (but see Adams et al., 1996). Also, acute SSRI treatment is both in rodents as well as in humans assocated with an acute increase in anxiolytic-like behaviour (Näslund et al., 2015), which is the opposite of the situation in zebrafish (Maximino et al., 2013; Karakaya et al., 2021; de Melo et al., 2019) and also at least another teleost (McDonald et al., 2011). Findings relating to cortisol responses are also divergent, with acute SRI treatment inducing a decrease in blood cortisol in zebrafish (Sander de Abreu et al., 2014) while an increase is typically seen in humans and other mammals (Hesketh et al., 2005; Nadeem et al, 2004). Also, both in rodents (Näslund et al., 2015) and in humans (Frick et al., 2015), there are reports of an association between high levels of serotonin/high capacity for serotonin release and/or production on one hand and high anxiety-like behaviour or anxiety disorders on the other hand, with treatment response in humans with generalised anxiety disorder being associated with a reduced capacity (Frick et al., 2016). Zebrafish data is divergent also here (strain differences in Maximino et al., 2013 but see Tran et al., 2016). As the authors mention, the acute serotonin elevation seen during early treatment is transient and the long-term effects are unclear; if anything the early rise in synaptic availability coincides with a period where patients experience heightened anxiety (e.g. Targum et al., 1989; Sinclair et al., 2009; Näslund et al., 2017). In short - zebrafish are pretty different from mammals in terms of how they respond to serotonergic interventions and this should be mentioned. I can't see how it would be better to use zebrafish than say, rats or tree shrews. At best it's neutral, but given these differences, the model organism is in itself a limitation of the paper.

Answer: I think that it is the main recommendation of the reviewer. So we markedly rewrote the Discussion in order to prove molecular and pharmacological similarities of the 5-HT system in zebrafish and mammals. At the same time, we discussed only the question concerning the role of 5-HT in anxiety in zebrafish and mice and the effects of antidepressants on anxiety-related behavior in these species. We added 5 new references clarifying this question. Of course, the information on the 5-HT mechanisms of anxiety in mammals is huge and very contradictory and its discussion is out of the present study. In the Discussion we provided three main points of similarity between zebrafish and rodents (lines 332-337).

There are, as the authors themselves discuss, indications of lowered serotonergic tone/transmission/availability after long-term treatment in rodents, and I don't really understand why inducing a hyposerotonergic state by way of p-CPA necessarily would be a model of treatment-resistant depression, as there are no indications from the clinical situation for that to be the case. Of course, the long-term effects of SSRI administration are unclear and there are a numerous hypotheses of their mode of action beyond the early effects at the serotonergic synapse, hypotheses that only partly overlap. In general, I feel that the discussion is far too focused on the situation in zebrafish for a paper that desires to in some way help in understanding a human phenomenon; "We showed that the combined administration of these drugs dramatically 256 reduced the 5-HT level in zebrafish brain more than pCPA and fluoxetine separately and, 257 therefore, we might expect an increase in “bottom dwelling” and normalization of lo- 258 comotor activity in zebrafish treated with these drugs. Instead we observed a paradoxical 259 decrease in “bottom dwelling” expression and locomotor activity." is instructive in this regard - it is paradoxical in terms of the zebrafish situation as increased 5-HT levels generally associate with less anxiety-like behaviour, but the opposite is the case in rodents as well as humans.

Answer: The main aim of our study was to model on zebrafish the effects of chronic antidepressant treatment on the brain and behavior in TPH2 deficient individuals (lines 66-68). We experimentally proved that TPH2 deficiency dramatically altered these effects of fluoxetine and pargyline and concluded that TPH2 deficiency could cause a refractory to antidepressant treatment (338-340). Although our results were obtained on zebrafish they revealed the general mechanisms common in fish and mammals (lines 332-337). We consider that the phrase “We showed that …” is unclear and it was corrected (lines 265-270).

Some shorter notes:

The acute effect of SSRIs in the forced-swim test is unlikely to be antidepressant in nature, considering the weeks needed for a therapeutic effect in humans, and in other tests/paradigms (e.g. chronic unpredictable stress) - see Anyan & Amir, 2018 for a good discussion of the subject.

Answer: We completely agree with this note. That is why in the present study we investigated effects of chronic, but not acute, treatment.

I would also strongly object to the statement that "According to commonly accepted neurotrophic hypothesis the therapeutic effect 41 of SSRIs is mediated by activation of cAMP-response protein (CREB) and synthesis of the brain derived neurotrophic factor (BDNF)" - it is one of many hypotheses, and it has rather little in terms of clinical/human data to support it (mostly some candidate gene studies as well as sparse and indirect post-mortem data, as far as I know)

.Answer: Of course, there are at least 5 main hypothesis of depressive disorders (5-HT, DA, NA, kynurenine and neurotrophic). Anxiety frequently accompanies depression. Since test for depressive-like behavior in zebrafish is developed, we used the novel tank diving test as a commonly accepted test for anxiety. Our study was designed to test 5-HT hypothesis of anxiety/depression.

There is also the conflation of models of depression and anxiety - it is not clear why the novel-tank diving test (or the scototaxis test) should be regarded as a model of the former state, and some discussion of what state in humans that the authors wish to model would be nice. This of course ties in to my general desire of clearer rationale for the paper.

Answer: Application of zebrafish to model the fundamental mechanism of psychic disorders and their pharmacological treatment is just beginning and now we still have no sufficient information for correct answer to the key philosophic questions of the reviewer. At the same time, we hope that our present study will add a brick in the future building of the domain of zebrafish neuroscience and psychopharmacology.

We thank the reviewer very much for his/her valuable comments.

Round 2

Reviewer 1 Report

Authors exhaustively clarified every point I toched in my revision.

Please correct sentence at line 214: "This result agrees with THAT/THOSE PREVIOUSLY observed earlier in zebrafish [33]"